# Therapeutic Potential of Hydrogen-Rich Water on Muscle Atrophy Caused by Immobilization in a Mouse Model

**DOI:** 10.3390/ph16101436

**Published:** 2023-10-10

**Authors:** Seyedeh Elnaz Nazari, Alex Tarnava, Nima Khalili-Tanha, Mahdieh Darroudi, Ghazaleh Khalili-Tanha, Amir Avan, Majid Khazaei, Tyler W. LeBaron

**Affiliations:** 1Metabolic Syndrome Research Center, Mashhad University of Medical Sciences, Mashhad 13131-99137, Iran; nazarie971@mums.ac.ir (S.E.N.);; 2Drink HRW, New Westminster, BC V3J 0B6, Canada; alextarnava@gmail.com; 3Department of Kinesiology and Outdoor Recreation, Southern Utah University, Cedar City, UT 84720, USA; 4Molecular Hydrogen Institute, Enoch, UT 84721, USA

**Keywords:** hydrogen-rich water, muscle atrophy, immobilization, oxidative stress

## Abstract

Skeletal muscle atrophy is associated with poor quality of life and disability. Thus, finding a new strategy for the prevention and treatment of skeletal muscle atrophy is very crucial. This study aimed to investigate the therapeutic potential of hydrogen-rich water (HRW) on muscle atrophy in a unilateral hind limb immobilization model. Thirty-six male Balb/C mice were divided into control (without immobilization), atrophy, and atrophy + hydrogen-rich water (HRW). Unilateral hind limb immobilization was induced using a splint for 7 days (atrophy) and removed for 10 days (recovery). At the end of each phase, gastrocnemius and soleus muscle weight, limb grip strength, skeletal muscle histopathology, muscle fiber size, cross-section area (CSA), serum troponin I and skeletal muscle IL-6, TNF-α and Malondialdehyde (MDA), and mRNA expression of NF-κB, BAX and Beclin-1 were evaluated. Muscle weight and limb grip strength in the H_2_-treated group were significantly improved during the atrophy phase, and this improvement continued during the recovery period. Treatment by HRW increased CSA and muscle fiber size and reduced muscle fibrosis, serum troponin I, IL-6, TNF-α and MDA which was more prominent in the atrophy phase. These data suggest that HRW could improve muscle atrophy in an immobilized condition and could be considered a new strategy during rehabilitation.

## 1. Introduction

Muscle atrophy, a complex physiological phenomenon characterized by the loss of skeletal muscle mass and strength, presents a significant challenge across various medical contexts. There are many circumstances, such as fractures, that lead to the immobilization of body parts which results in muscle atrophy. The maintenance of skeletal muscle mass is very important not only because of its effects on quality of life, or increasing morbidity or mortality, but it also has adverse socioeconomic ramifications [1,2,3]. The preservation of skeletal muscle mass is pivotal not only for enhancing quality of life but also for reducing morbidity and mortality rates. As such, there exists an urgent need to comprehensively understand the intricate mechanisms underpinning muscle catabolism during periods of unloading or immobilization.

Despite extensive research, the precise molecular pathways governing muscle atrophy in response to unloading remain elusive. This knowledge gap hinders the development of effective therapeutic interventions aimed at ameliorating muscle wasting. Currently, exercise and physical therapy stand as the foremost strategies to counteract muscle loss and reinstate skeletal muscle mass. However, the applicability of these approaches is limited in certain clinical scenarios, such as patients with fractures. Consequently, a critical unmet need persists for novel interventions capable of mitigating muscle atrophy in immobilization settings.

A prevailing hypothesis posits several etiological mechanisms contributing to the onset and progression of muscle atrophy during immobilization. Among these, heightened oxidative stress within inactive muscles emerges as a prominent candidate. Accumulating evidence underscores the robust generation of reactive oxygen species (ROS) within skeletal muscles during limb immobilization and prolonged inactivity [4,5]. Moreover, a lack of exercise and movement results in a decreased hormetic response leading to lower endogenous antioxidant status, which further contributes to oxidative stress [6]. In addition, oxidative stress has been shown to participate in the activation of proteolysis via different mechanisms, all of which are activated during disuse muscle atrophy [7].

In this intricate landscape of muscle atrophy, molecular hydrogen emerges as a potentially transformative therapeutic modality. Demonstrating multifaceted attributes, molecular hydrogen acts as an antioxidant and anti-inflammatory agent, functioning as a novel medical gas [8]. Additionally, it helps regulate autophagy, prevents premature apoptosis, and protects against various forms of cellular stresses (e.g., radiation, hypoxia, ROS, toxins, etc.) [9]. Molecular hydrogen can be administered via inhalation or simply dissolved into water and ingested [9]. Numerous pieces of evidence indicate that hydrogen-rich water (HRW) has anti-inflammatory and antioxidative properties which uniquely qualify it as a potential candidate as a novel therapy to suppress the development of diseases caused by oxidative stress, cardiovascular disease, cancer, diabetes, and metabolic disease [10,11,12].

In skeletal muscle, HRW could reduce mitochondrial oxidative stress and inflammation after eccentric exercise [13]. Also, intermittent HRW could reduce lactate production, muscle soreness, and improve muscle function [12]. Mechanistically, molecular hydrogen has been shown to activate the Nrf2/Keap1 (nuclear factor erythroid related factor-2 [Nrf2])–Keap1 (Kelch-like erythroid cell-derived protein with CNC homology [ECH]-associated protein 1) antioxidant signaling pathway, thereby increasing endogenous antioxidants [14]. It has been proposed that the H_2_ molecule may act as an exercise mimetic due to its redox adaptogenic properties and activation of similar pathways as exercise [15].

Against this backdrop, the current study endeavors to extend the understanding of hydrogen-rich water’s potential therapeutic effects on muscle atrophy. Specifically, we seek to examine the impact of HRW during both the atrophy and recovery phases in a murine model of unilateral hind-limb immobilization. By comprehensively elucidating the influence of HRW on the multiple facets of muscle atrophy, our investigation aims to contribute substantively to the development of novel strategies for counteracting muscle wasting in immobilization scenarios.

## 2. Results

### 2.1. Morphological Changes and Weight of Muscles

Limbs were immobilized for seven days, the splints were removed and in half of the animals in each group (i.e., “atrophy” group and atrophy + H_2_; “H_2_” group) the affected limbs were observed macroscopically. Administration of control water or HRW was continued for another 10 days (recovery phase) in the remaining half of the animals in each group (Figure 1A). During the atrophy phase, our results showed that the gastrocnemius muscles of the atrophy group had significantly less weight than the control, while treatment by HRW significantly improved muscle weight (Figure 1(B-Left) and Figure 2A–C). The recovery period was continued in half of the animals in each group (i.e., atrophy and atrophy + H_2_) and after 10 days of opening the splint, the muscle weight was measured again and normalized to body weight. The results showed that atrophied muscle in atrophy and H_2_ groups significantly improved (Figure 1(B-Right) and Figure 2A–C). Gastrocnemius muscle weight was significantly higher in the H_2_ group than in the atrophy group. There was no significant difference in the sum weight of gastrocnemius and soleus muscles in three groups between the atrophy and recovery phases.

### 2.2. Four-Limb Grip Strength

A four-limb strength test is a simple non-invasive method for the evaluation of mouse muscle force. We measured limb strength three times and mean strength was reported as absolute strength change and calculated as the percentage of the measurements performed at the end of the study (day 18) with respect to the time of splint removal (day 8) and day 0. Figure 3 illustrates absolute strength over the time period and is also normalized to body weight (strength/body weight). To determine if HRW could improve muscle strength, the grip strength at the end of the 7-day immobilization period was measured. It was found that grip strength decreased by 30.33 ± 3.28% in the atrophy group, whereas the control group slightly increased (6.63 ± 5.37%). However, in contrast to the atrophy group, grip strength only decreased by 15.02 ± 5.5% in the H_2_-treatment group. Moreover, after the 10-day recovery period, forelimb grip strength increased in the control group (12.87 ± 12.44%) but was still reduced by 16.01 ± 3.61% than in the control group. However, grip strength increased by 1 ± 3.71% in the H_2_-treatment group (Figure 3E).

### 2.3. Histological Findings

Representative images of H&E-stained sections of gastrocnemius muscles are shown in Figure 4. At day 7 post-immobilization (Figure 4A) and 10 days after removing the splint in the recovery phase (Figure 4B); there is a significant improvement in myofiber diameter in the skeletal muscles of the H_2_-treated group. Figure 4C,D illustrates the cross-section area and distribution graph of fiber size from the immobilized limb over the 7-day course in all experimental groups. Our results indicated that the relative frequency of muscle fiber size in the atrophy group was between 500 to 1250 μm^2^ and most of them corresponded to 750 to 1000 μm^2^ which was significantly lower than the control group (most fiber size was between 1750–2000 μm^2^) and HRW treatment significantly increased muscle fiber size during immobilization. Evaluation of the cross-section area in histological samples indicated a lower cross-section area in the atrophy group than control which was significantly improved by HRW treatment (Figure 4D).

### 2.4. Biochemical Measurements

Results showed that after 7 days of immobilization, serum troponin I levels in the atrophy group were significantly increased compared to the control and that H_2_ treatment reduced it (Figure 5A). After removing the splint at the end of the recovery phase, serum troponin I was reduced in the atrophy group by day 10 of recovery to near control levels. Thus, there were no significant differences in troponin I serum levels in the three study groups (Figure 5A).

The MDA levels, a marker of oxidative stress, in skeletal muscle tissue were increased after immobilization. However, HRW treatment significantly attenuated this increase to near control levels (Figure 5B). Comparing the tissue level of MDA in atrophy and recovery phase indicates that treatment by HRW significantly further reduced MDA level at the end of the recovery phase which reached the level of the control group. Skeletal muscle tissue levels of IL-6 and TNF-α in the atrophy group were higher than control, and treatment with HRW significantly prevented the rise in tissue levels of these inflammatory markers compared to the atrophy group.

### 2.5. Gene Expression Analysis

To investigate the inflammatory biomarkers (IL-6, TNF-α, and NF-κB) and atrophy mediators (Bax, Beclin 1), qRT-PCR was used to analyze mRNA levels in the gastrocnemius muscle. The results showed that, after 7-days of immobilization, the total expression of inflammatory biomarkers of muscle tissues (IL-6, TNF-α) was significantly reduced in the H_2_-treatment group compared to the atrophy group (*p* < 0.05). The expression levels of NF-κB and Beclin 1 were significantly increased in the treatment group in comparison with the atrophy group (*p* < 0.01). Considering that the expression of the Bax gene increases in the atrophy group, Bax was also over-expressed in the H_2_-treatment group (Figure 6A–C).

## 3. Discussion

Understanding the mechanism of skeletal muscle atrophy is essential for developing a new strategy for the prevention or treatment of skeletal muscle atrophy [2]. Inflammation and oxidative stress play contributing roles in the etiology of muscular atrophy. Nevertheless, conventional antioxidants or anti-inflammatories do not have the desired protective effects. Molecular hydrogen has been demonstrated to have unique and selective antioxidant and anti-inflammatory effects by modulating these cellular pathways [9]. In this study, we found that treatment by HRW, especially during immobilization, could improve muscle atrophy and restore muscle mass and strength.

Muscle atrophy was obviously induced following seven days of immobilization including localized fibrosis at the site of injury, which corresponds to what has been observed in previously published papers [16,17,18]. However, treatment with HRW attenuated the loss of muscle weight and grip strength of immobilized hind limbs compared to the atrophy-only group. Previous studies have demonstrated that administration of intermittent HRW improves muscle function and alleviates muscle soreness with an accompanying reduction in lactate levels [13]. Other studies indicated that HRW attenuates muscular damage and improves sore muscles after high-intensity eccentric exercise [19,20]. We showed that HRW increases muscle fiber size and reduces muscle fibrosis which is substantially needed for improving muscle grip strength. The antioxidative and anti-inflammatory effects of H_2_ have been documented [21]. Thus, it seems that H_2_ could effectively mitigate atrophy-induced loss of muscle mass and muscle strength by reducing oxidative stress, inflammation, and apoptosis [22].

For example, our results showed higher tissue MDA levels in atrophied muscles compared to those treated with HRW. Disruption of redox signaling, due to increased production of reactive oxygen species (ROS), is an important regulator of signaling pathways that control both proteolysis and protein synthesis in skeletal muscle [23]. Importantly, molecular hydrogen acts as a selective antioxidant and does not neutralize important signaling ROS (e.g., hydrogen peroxide, superoxide, nitric oxide, etc.). The H_2_ molecule can only react with and neutralize the toxic hydroxyl radical and to a lesser extent peroxynitrite [15]. Moreover, it has recently been demonstrated that H_2_ can interact with the Fe-porphyrin molecule [24], which is rich in mitochondria and myoglobin and thus rich in type I muscle fibers. Perhaps this makes these muscle fibers another ideal tissue for H_2_ to exert its favorable therapeutic effects. However, the effects of molecular hydrogen on muscle tissue have not been extensively evaluated. Several studies indirectly asses this by evaluating the effects of molecular hydrogen on exercise performance, which clinical studies have demonstrated reduced lactate production [25] and levels of fatigue [26]. Animal studies demonstrate that H_2_ treatment attenuated the exercise-induced increases in skeletal muscle TNF-α, NF-κB, IL-6, and caspase-3 levels while increasing superoxide dismutase levels [20]. Moreover, another study found that compared to but unlike vitamin C, H_2_ treatment did not reduce the exercise-induced elevated levels of peroxisome proliferator-activated receptor-gamma (PPARγ) coactivator-1alpha (PGC-1α), nuclear transcription factors 1–2 (NRF-1,2), and mitochondrial transcription factor A (TFAM) [27]. These studies indicate that H_2_ can reduce excessive levels of inflammation and oxidative stress, but does not interfere with favorable exercise-induced adaptations in the skeletal muscle. Importantly, unless there is some form of stressor, the biological effects of H_2_ are often not observed, thus it has no obvious effect on healthy muscle tissue [28,29].

It is indicated that inflammatory cytokines such as TNF-α could mediate ROS production which is involved in muscle wasting [30]. Therefore, it seems that HRW could improve muscle atrophy by reducing the expression and production of inflammatory markers and subsequently MDA production. Indeed, we also demonstrated that, in addition to reduced MDA levels, the skeletal muscle expression levels of the inflammatory markers, IL-6 and TNF-α, were significantly decreased in the H_2_-treated group. Studies indicated that inflammatory cytokines such as TNF-α and IL-6 have a critical role in the loss of skeletal muscle mass, especially in chronic diseases [31]. In addition, proinflammatory cytokine TNF-like weak inducer of apoptosis (TWEAK) has a role in muscle wasting and could degrade myosin heavy chain [32]. We found that levels of TNF-α and IL-6 were decreased by HRW, which may help explain the attenuation of muscle loss.

Interestingly, we also observed that NF-κB was upregulated in the HRW group. It might be expected that NF-κB would be lower since its activation can also be induced by TNF-α, which was reduced in our study. This may suggest that H_2_ modulated NF-κB via the non-canonical signaling pathway [33]. Furthermore, numerous studies on molecular hydrogen demonstrate that the cytoprotective effect of H_2_ is associated (mediated) by the suppression of NF-κB activation. Nevertheless, there are several studies that demonstrate that the cytoprotective effects of H_2_ were mediated by an initial transient upregulation of NF-κB followed by its decrease [34,35], which may be a form of hormesis [36].

On the one hand, increased activation of NF-κB has been observed in different types of skeletal muscle atrophy [37], as it contributes to skeletal muscle wasting in mice, and inhibition of NF-κB improves muscle atrophy [38]. On the other hand, since NF-κB is a proinflammatory transcription factor that regulates the expression of some genes that are involved in muscle proteolysis and fibrosis [30], increased NF-κB levels may be cytoprotective. It is indicated that one function of NF-κB is promoting cell survival and preventing cell death, and NF-κB induces the expression of different anti-apoptotic genes [37,39].

Although in the present study, we did not evaluate apoptosis in immobilized muscle tissue, we did show that the expression of two anti-apoptotic genes, BAX and beclin-1, increased after HRW treatment which is supported by previous studies [40,41]. Previous evidence reported that high expression of BAX and beclin-1-induced autophagy and reduced apoptosis can improve atrophy [42]. Molecular hydrogen has also been demonstrated to influence autophagy via a variety of mechanisms [22]. Importantly, hydrogen does not appear to merely enhance or suppress autophagy but appears to have a regulatory effect. In some cases, hydrogen may exert therapeutic effects via upregulating autophagy, whereas in other scenarios it may suppress excessive autophagy [9]. Hydrogen-induced autophagy regulation may be an important mechanistic contributor to the favorable effects on muscle atrophy. Recent findings revealed that the autophagy–lysosome system not only eliminates toxic molecules and dysfunctional organelles but also emerges as a critical system in myofiber survival [43].

Despite these favorable findings on muscle atrophy, the underlying mechanistic action of molecular hydrogen on atrophy attenuation remains elusive. A better understanding of the etiology of muscular atrophy is needed to determine the actions of molecular hydrogen. Furthermore, although our animal model of muscular atrophy demonstrates favorable therapeutic effects, clinical research is lacking. Due to the simplicity and safety of molecular hydrogen, clinical research on the effects of molecular hydrogen on muscle atrophy is warranted.

## 4. Materials and Methods

### 4.1. Materials

Balb/C mice (age: 10–12 weeks, weight: 24–26 g) purchased from Pasteur Institute of Iran. Hydrogen-producing tablets were obtained from HRW Natural Health Products Inc., New Westminster BC, Canada. The tablets generate molecular hydrogen upon reacting with the water according to the reaction Mg (s) + 2H^+^ (aq) → Mg^2+^ (aq) + H_2_ (g). The acid (H^+^) comes from organic acids, malic acid, and tartaric acid. ELISA kits were purchased from ZellBio Co. (ZellBio GmbH, Ulm, Germany). Force meter was purchased from IMADA Co. (Tomayashe, Japan). Other materials were purchased from Sigma Co. (Saint Louis, MO, USA).

### 4.2. Animals

Male Balb/C mice were kept under standard laboratory conditions with a 12 h light/dark cycle. The animals had free access to water and food ad libitum. All experimental procedures followed the Guide for the Care and Use of Laboratory Animals of Mashhad University of Medical Sciences and were approved by the local ethical committee (Ethich no and approval date: 4000877; 3 November 2021). The animals were anesthetized with an intraperitoneal (IP) injection of ketamine (100 mg/kg) and xylazine (10 mg/kg) prior to immobilization. At the end of the experiment, the animals were sacrificed by cervical dislocation, and the gastrocnemius and soleus muscles were removed.

### 4.3. Immobilization Atrophy and Muscle Recovery

For induction of atrophy, mice were exposed to a non-invasive model of unilateral hind limb immobilization as previously described [44]. Briefly, the right hind limb was shaved and a splint, made using a capless 1.5 mL microfuge tube, was used for unilateral mouse hind limb immobilization (Figure 1A). For this purpose, the right hind limb was inserted into the cylindrical part of the splint and wrapped with the knee in an extended, and the ankle in a plantar-flexed position, for seven days (atrophy phase). Then, the splint was removed to evaluate muscle recovery after ten consecutive days (recovery phase). The procedure was performed using mouse restrainers without anesthesia. Age-matched non-immobilized mice with normal physical activity were used as the control group.

### 4.4. Animal Groups and Experimental Design

Thirty-six Balb/C male mice were randomly divided into three groups (*n* = 12 each): control, immobilized group (atrophy), and atrophy + HRW-treated group (H_2_). During the atrophy phase, the H_2_ group received HRW which was prepared by dissolving one hydrogen-producing tablet in usual drinking water (500 mL bottle), twice daily every 12 h. The initial concentration of hydrogen water was more than 1.5 mM and remained >0.1 mM by the end of the 12 h interval as measured by redox titration (H2Blue^TM^; H2 Sciences, Las Vegas, NV, USA) as described previously [45,46]. Control and atrophy groups received normal drinking water. The average daily water consumption for each mouse in all groups was approximately 15 mL. On day 8, half of the animals in each group were sacrificed for sample collection. The remaining half in each group had their splints removed and continued to receive either HRW or control water for the next 10 days (recovery phase).

### 4.5. Limb Strength Grip Test

Limb strength was determined on day 0, day 7 (end of immobilization), and day 18 (end of recovery time). Limb strength was measured using a digital force gauge. For this purpose, the animals grasped a mesh and the mouse’s tail was pulled directly toward the force meter until the hind limbs were released from the mesh assembly. The mean value of three sessions was measured and calculated as the percentage of the measurements with respect to the baseline (day 0) and normalized to body weight. A single operator performed this procedure to reduce variability.

### 4.6. Sample Collection

The animals were sacrificed at the end of the atrophy and recovery phases. Blood samples were taken, and gastrocnemius and soleus muscles were harvested. The gastrocnemius muscles were divided into three equal parts and the proximal part of the samples were frozen in liquid nitrogen (LN) to be stored at −80 °C to determine molecular and biochemical parameters further. Another part of the gastrocnemius muscles was put in formalin 10% for histological evaluations. Serum Aliquots were also held at −20 °C to be used for evaluation of troponin I concentration.

### 4.7. Histological Evaluation

Paraffin-embedded muscle samples were prepared at a thickness of 5 µm and stained with Hematoxylin-Eosin (H&E) for evaluation of fiber size and cross-section. Five different images of muscle cross-sections at a 10× magnification were taken and analyzed using NIH Image J software (Version: 1.53t).

### 4.8. Serum and Tissue Measurements

Skeletal muscle IL-6 and TNF-α levels, and serum troponin I levels were assessed using specific sandwich ELISA kits according to the manufacturer’s instructions. Malondialdehyde (MDA) content of the muscle was measured using the method previously described [46]. Briefly, 2 mL of reagent solution containing TBA, TCA, and HCL was mixed with 1 mL supernatant fluid and exposed to the hot water (100 °C) for one hour and centrifuged for five min at 4000–5000 rpm. MDA content was calculated by the following equation (where C (M): concentration in molar, A: optical density): C (M) = A/1.56 × 105.

### 4.9. Extraction of RNA and Quantitative Real-Time RT-PCR

Relative expression of the inflammatory biomarkers (IL-6, TNF-α, NF-κB) and apoptotic mediators (BAX, Beclin 1) was determined using specific qPCR Primers for target genes (Macrogene Co., Seoul, Republic of Korea). RNA extraction was performed immediately after homogenizing the gastrocnemius muscle using an RNeasy Mini kit (Parstoos, Mashhad, Iran). A NanoDrop detector was used to determine the quantity and purity of RNA isolated (Nanodrop Technology-1000, NanoDrop Technologies LLC, Wilmington, DE, USA). cDNA was reverse transcribed from the extracted RNA by the cDNA synthesis kit (CinaColon, Tehran, Iran). On a light cycler (Roche, Basel, Switzerland) we used SYBER Green qPCR Master mix to perform all the PCR reactions. A comparative method was used to analyze the mRNA level normalized against GAPDH (2^−ΔΔCt^). The PCR primer sequences are presented in Table 1.

### 4.10. Statistical Analysis

Data were reported as mean ± standard error (SE) and analyzed by SPSS 20 software. The differences between groups were analyzed using One-way analysis of variance (ANOVA) followed by Tukey’s Posthoc. *p* less than 0.05 was considered statistically significant.

## 5. Conclusions

Our results, in conjunction with other studies, indicate that HRW has anti-inflammatory and antioxidant effects. These benefits may mediate the protective effects of molecular hydrogen against muscle atrophy. Furthermore, H_2_ may have some hormetic effects due to the increased NF-κB signaling, as indicated by previous studies. Our results indicate that the administration of HRW can be considered as a new strategy for the prevention and restoration of skeletal muscle mass to immobilization. However, more mechanistic and clinical studies are needed to elucidate the molecular effects and confirm the viability of this novel treatment.

## Figures and Tables

**Figure 1 pharmaceuticals-16-01436-f001:**
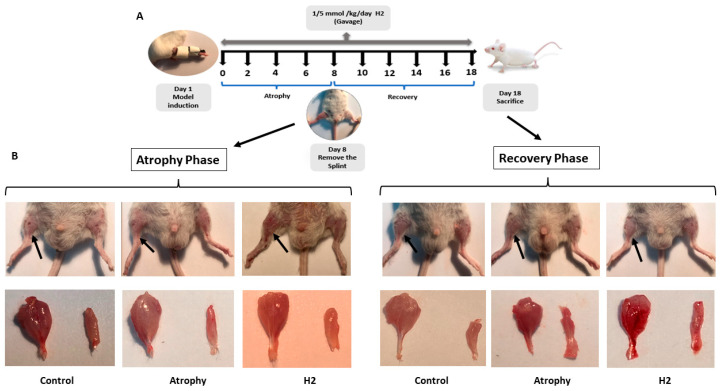
Schematic representation of the timeline of the experimental protocol (**A**). Macroscopic images of the hind limb and gastrocnemius-soleus muscles (indicated by black arrows) at the end of the atrophy phase (**B**-**Left**) and recovery period (**B**-**Right**).

**Figure 2 pharmaceuticals-16-01436-f002:**
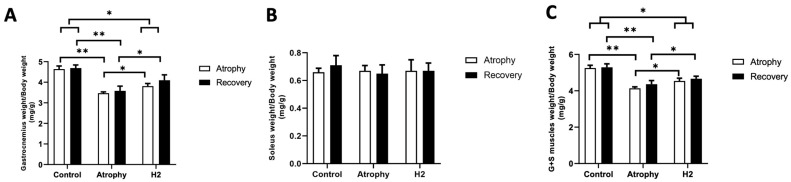
Effect of treatment by HRW on muscle atrophy induced by 7-day immobilization ((**A**–**C**) “Atrophy”), and after removal of splint (at the end of recovery period) ((**A**–**C**) “Recovery”). * *p* < 0.05, ** *p* < 0.01. G + S: gastrocnemius + soleus muscles; *n* = 6 for each group.

**Figure 3 pharmaceuticals-16-01436-f003:**
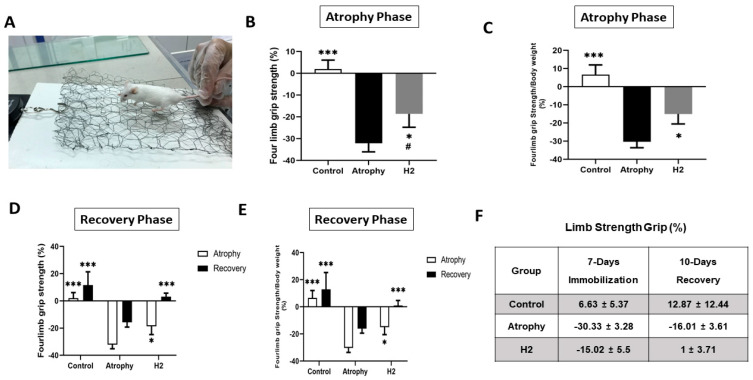
For measurement of grip strength, the animals grasped a mesh and the mouse’s tail was pulled directly toward the force meter until the hind limbs were released from the mesh assembly (**A**). Four limb strength and normalized strength in atrophy phase (**B**,**C**) and recovery phase (**D**,**E**). (**F**): percent of limb grip strength after 7-day immobilization and 10-day immobilization. Data were presented as mean ± standard error (Mean ± SE). (*** *p* < 0.001, * *p* < 0.05 compared to the atrophy group. # *p* < 0.05 compared to the control group); *n* = 6 for each group.

**Figure 4 pharmaceuticals-16-01436-f004:**
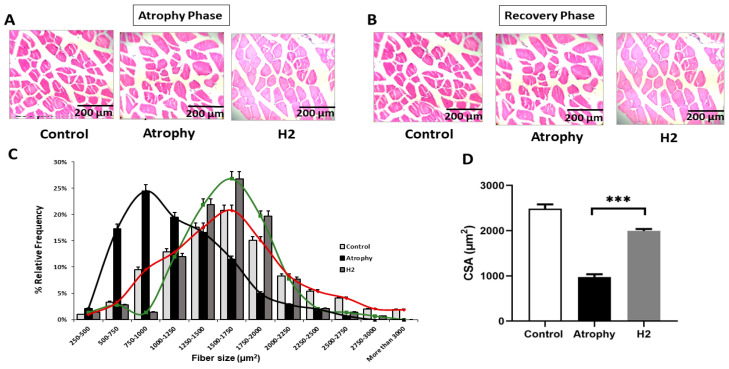
Representative image of H&E-stained sections of gastrocnemius muscles at the end of atrophy (**A**) and recovery phase (**B**) (×100). Quantitative assessment of muscle fiber size (**C**) and mean cross-section area of each muscle fiber at the end of the atrophy phase; Black: Atrophy; Green: Control; Red: H2 group (**D**). Cross-sectional area was measured and calculated from 150 skeletal muscle fibers in H&E-stained sections using Image J software (Version: 1.53t). (*** *p* < 0.01 compared to the atrophy group); *n* = 6 for each group.

**Figure 5 pharmaceuticals-16-01436-f005:**
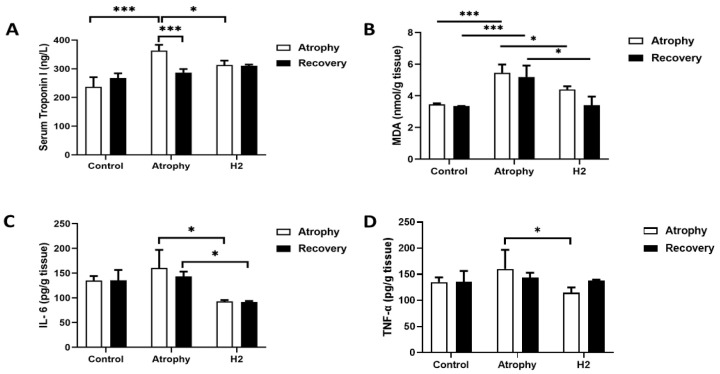
Effects of H_2_ treatment on serum troponin I (**A**) and skeletal muscle MDA (**B**), IL-6 (**C**), and TNF-alpha (**D**). (*** *p* < 0.01, * *p* < 0.05 compared to the atrophy group); *n* = 6 for each group.

**Figure 6 pharmaceuticals-16-01436-f006:**
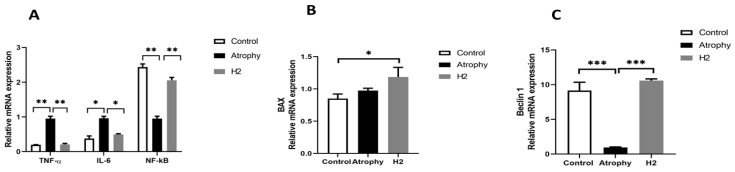
Effect of 10-day immobilization in atrophy and H_2_-treated group on mRNA expression of inflammatory markers (**A**), BAX (**B**), and Beclin (**C**). H_2_ treatment resulted in downregulation of inflammatory markers (TNF-α and IL-6) and upregulation of NF-κB, BX, and Beclin-1. (*** *p* < 0.01, ** *p* < 0.01, * *p* < 0.05 compared to the atrophy group); *n* = 6 for each group.

**Table 1 pharmaceuticals-16-01436-t001:** The sequence of primers.

Gene	Source	Primer	Sequence (5′ to 3′)
GAPDH	Mouse	Forward	CAACGACCCCTTCATTGACC
		Reverse	CTTCCCATTCTCGGCCTTGA
NF-KB	Mouse	Forward	CCAGCTTCCGTGTTTGTTCA
		Reverse	AGGGTTTCGGTTCACTAGTTTCC
Beclin 1	Mouse	Forward	ATTTCAGACTGGGTCGCTTG
		Reverse	TTATTGGCCAAAGCATGGAG
Bax	Mouse	Forward	AGACAGGGGCCTTTTTGCTAC
		Reverse	AATTCGCCGGAGACACTCG
IL-6	Human	Forward	TCTGGAGCCCACCAAGAACGA
		Reverse	TTGTCACCAGCATCAGTCCCA
TNF-α	Mouse	Forward	AGGCTGTCGCTACATCACTG
		Reverse	CTCTCAATGACCCGTAGGGC

## Data Availability

Data is contained within the article.

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
