# Peer review of "Therapeutic Potential of Hydrogen-Rich Water on Muscle Atrophy Caused by Immobilization in a Mouse Model"

_pharmaceuticals, 2023, doi:10.3390/ph16101436_

Round 1

Reviewer 1 Report (Previous Reviewer 1)

The manuscript describes a study to investigate the effect of hydrogen-rich water (HRW) on muscle atrophy during the atrophy and recovery phases in a mouse model with unilateral hindlimb immobilization. The current state of knowledge does not allow to pinpoint the exact causes of muscle catabolism during immobilization or load. Molecular hydrogen used as a medical gas has been shown to have antioxidant, anti-inflammatory effects, help regulate autophagy, prevent premature apoptosis, and protect against various forms of cellular stress. Therefore, the goal presented by the authors is in line with the trend of modern medical research.

After a careful reading of the manuscript, I conclude that the abstract, introduction and remaining chapters comprehensively and adequately capture the issues discussed. The conclusions presented by the authors are consistent with the evidence and address the main research problem. In addition, I believe that the selection of bibliographic items is appropriate.

 I recommend publication in its present form.

Quality of English Language is adequately good

Author Response

Thank you for reviewing our manuscript and positive feedback. 

Reviewer 2 Report (Previous Reviewer 2)

The authors did a good job in the improvement of the language of the manuscript and addressing most of my comments.

Nevertheless, my critiques of Figure 5 were not addressed. I strongly suggest the removal of Figure 5 from the manuscript. In its present form, the data presented in Figure 5 cannot be trusted. It shows/measures in the longitudinal sections the collagen in the perimysium/epimysium. This is not what changes with fibrosis or is important for the immobilization-induced impairments in muscle function.

What should have been measured is changes in the collagen composition in the endomysium in the high-resolution images. This was measured in references 17-18 the study listed in the Discussion. Unless the authors can provide data similar to the presented in references 17-18, Figure 5 should not be used in the manuscript.

Please also see the reference below for the example of proper measurements of muscle fibrosis:

Natsumi Tanaka MS, Yuichiro Honda PhD, Yasuhiro Kajiwara MS, Hideki Kataoka PhD, Tomoki Origuchi MD, Junya Sakamoto PhD, Minoru Okita PhD. 2022. Myonuclear apoptosis via cleaved caspase-3 upregulation is related to macrophage accumulation underlying immobilization-induced muscle fibrosis. Muscle &Nerve 65 (3): 341-349.

Discussion

Line 216: “which is rich in mitochondria and thus rich in muscles”. Only slow muscle fibers are rich in mitochondria. Fast glycolytic fibers are mitochondria-poor.

Author Response

Reviewer 1:

The authors did a good job in the improvement of the language of the manuscript and addressing most of my comments. 

Thank you

Nevertheless, my critiques of Figure 5 were not addressed. I strongly suggest the removal of Figure 5 from the manuscript. In its present form, the data presented in Figure 5 cannot be trusted. It shows/measures in the longitudinal sections the collagen in the perimysium/epimysium. This is not what changes with fibrosis or is important for the immobilization-induced impairments in muscle function.What should have been measured is changes in the collagen composition in the endomysium in the high-resolution images. This was measured in references 17-18 the study listed in the Discussion. Unless the authors can provide data similar to the presented in references 17-18, Figure 5 should not be used in the manuscript.Please also see the reference below for the example of proper measurements of muscle fibrosis:

Natsumi Tanaka MS, Yuichiro Honda PhD, Yasuhiro Kajiwara MS, Hideki Kataoka PhD, Tomoki Origuchi MD, Junya Sakamoto PhD, Minoru Okita PhD. 2022. Myonuclear apoptosis via cleaved caspase-3 upregulation is related to macrophage accumulation underlying immobilization-induced muscle fibrosis. Muscle &Nerve 65 (3): 341-349.

Reply: Thank you for your response. We deleted figure 5 and related sentences and corrected the figures number…

Discussion 

Line 216: “which is rich in mitochondria and thus rich in muscles”. Only slow muscle fibers are rich in mitochondria. Fast glycolytic fibers are mitochondria-poor.

Reply: That is correct. We updated our statement, which also incorrectly suggested that muscles might be a better target than other tissues with even higher numbers of mitochondria. The point here is to illustrate that muscles, specifically type I, could also be a site of interaction of H2 due to the presence of the Fe-porphyrin molecule.

Reviewer 3 Report (Previous Reviewer 3)

The current manuscript entitled ‘Therapeutic potential of hydrogen-rich water on muscle atrophy caused by immobilization in a mouse model’ has been improved compared the previous version, however there are still some crucial points to be addressed.

I have concerns about the experimental groups. 

Firstly, the use of both the terms Atrophy phase and Atrophy group is misleading. If I understood properly, the Atrophy group received normal drinking water, I suggest to replace the name of this group.

Secondly, I think that it is not clear what are the groups in the recovery phase: is the group named Atrophy in the recovery phase free of the splint or not (lines 90-91 are confusing)?

Why does the control (see for example fig2 panels D,E,F) display an atrophy column?

Author should carefully describe the groups in paragraph 4.4, and in Fig 1.

In line with the previous point, the meaning of the two sets of graphs in figure 2 and 3 is not clear to me. Are the graphs reporting the recovery phase, also including the atrophy phase data (so repeating the values of panels A,B,C in fig 2 and A,B in fig 3)? Or, are they reporting animals with or without splint?

Accordingly, in Figure 6 there is only one set of graphs. Do they correspond to the same experimental conditions of the panels D,E and F of figure 2?

Histological specimens and some markers (see fig 7) have been analyzed only on atrophy phase samples, is it possible to perform the analyses on the recovery phase samples?

Authors should report in materials and methods how the percentage of fibrosis, with Masson staining, was determined. I find puzzling that the control group presents no staining, which normally should be present (see for reference: Diantha Van De Vlekkert, Eda Machado, and Alessandra d’Azzo, 2020).

The graphs in the figures should be more homogeneous in shape and characteristics, the statistical significance should be indicated in the same way in the distinct figures. In Fig 2 panel D and F have two asterisks on the control.

In the legend of figure 2 Authors should indicate the abbreviations of G+S.

Authors should indicate also the number of samples analyzed in the legends.

Authors did not test the effect of H2 on healthy skeletal muscles, can supplemental information be retrieved from literature? Can Authors speculate that, as described for other antioxidants, H2 exerts its action only in the presence of damage?

Moderate editing

Author Response

The current manuscript entitled ‘Therapeutic potential of hydrogen-rich water on muscle atrophy caused by immobilization in a mouse model’ has been improved compared the previous version, however there are still some crucial points to be addressed. I have concerns about the experimental groups. 

Thank you

Firstly, the use of both the terms Atrophy phase and Atrophy group is misleading. If I understood properly, the Atrophy group received normal drinking water, I suggest to replace the name of this group. Secondly, I think that it is not clear what are the groups in the recovery phase: is the group named Atrophy in the recovery phase free of the splint or not (lines 90-91 are confusing)?Why does the control (see for example fig2 panels D,E,F) display an atrophy column?

Reply: Fig 2 was updated extensively, and a clearer description was added. The atrophy column indicates the muscle weight of the atrophy only group, which group is compared to the atrophy + H2 group (designated as “H2”) and the control group.

Author should carefully describe the groups in paragraph 4.4, and in Fig 1. In line with the previous point, the meaning of the two sets of graphs in figure 2 and 3 is not clear to me. Are the graphs reporting the recovery phase, also including the atrophy phase data (so repeating the values of panels A,B,C in fig 2 and A,B in fig 3)? Or, are they reporting animals with or without splint? Accordingly, in Figure 6 there is only one set of graphs. Do they correspond to the same experimental conditions of the panels D,E and F of figure 2?

Reply: Fig 2 was updated extensively, and a clearer description was added. We show Fig 1 A to help illustrate the study design. We clarified the groups in the Methods section 4.4. In short, we had two phases: atrophy phase and recovery phase. We had a control group (No splint, No treatment), Atrophy group (Splint, No treatment), and H2 (Splint + treatment) (n=12 each). After 8 days (end of the atrophy period), half of the animals in each group were sacrificed and samples were taken, and in the rest, splints were removed from atrophy and H2 groups for 10 days (recovery period) and in this phase treatment with HRW was continued in H2 group to see if HRW could improve muscle atrophy more compared to atrophy group without treatment. Thus, for example, fig 2 explains the gastrocnemius muscle and soleus muscle weight in three experimental groups at the end of atrophy phase (Fig 2 A-C in white columns designated “atrophy”) and at the end of recovery period (Fig. 2 A-C in solid/black columns designated as “Recovery”), showing more atrophy improvement in H2 treatment compared to control or atrophy group without treatment especially at the end of atrophy phase.

Histological specimens and some markers (see fig 7) have been analyzed only on atrophy phase samples, is it possible to perform the analyses on the recovery phase samples?

Reply: Since the HRW treatment did not have significant effect in recovery phase and improvement in muscle atrophy during recovery phase was not significant compared to atrophy group, we didn’t more experiment in this phase. Thus, we did more experiments at the end of atrophy phase to find the mechanism and see changes of mRNA expression of inflammatory and apoptotic/antiapoptotic factors.  

Authors should report in materials and methods how the percentage of fibrosis, with Masson staining, was determined. I find puzzling that the control group presents no staining, which normally should be present (see for reference: Diantha Van De Vlekkert, Eda Machado, and Alessandra d’Azzo2020).

Reply: we deleted the figure 5 based on the other reviewer's comment.

The graphs in the figures should be more homogeneous in shape and characteristics, the statistical significance should be indicated in the same way in the distinct figures. In Fig 2 panel D and F have two asterisks on the control.

Reply: We revised Fig. 2.

In the legend of figure 2 Authors should indicate the abbreviations of G+S.

Reply: We added (highlighted in green).

Authors should indicate also the number of samples analyzed in the legends.

Reply: we corrected. (Highlighted in yellow)

Authors did not test the effect of H2 on healthy skeletal muscles, can supplemental information be retrieved from literature? Can Authors speculate that, as described for other antioxidants, H2 exerts its action only in the presence of damage?

Reply: Thank you for the question. It appears that unless there is a direct stressor on the muscle tissue, that H2 has no obvious effect. We added a discussion on this as also compared to vitamin C. There is a paucity of direct research on this topic.

Round 2

Reviewer 2 Report (Previous Reviewer 2)

In the revised manuscript the authors addressed all of my critiques.

Reviewer 3 Report (Previous Reviewer 3)

The Authors have addressed all the points raised in the revision.

Moderate editing is required.

This manuscript is a resubmission of an earlier submission. The following is a list of the peer review reports and author responses from that submission.

Round 1

Reviewer 1 Report

The manuscript describes a study to investigate the effect of hydrogen-rich water (HRW) on muscle atrophy during the atrophy and recovery phases in a mouse model with unilateral hindlimb immobilization. The current state of knowledge does not allow to pinpoint the exact causes of muscle catabolism during immobilization or load. Molecular hydrogen used as a medical gas has been shown to have antioxidant, anti-inflammatory effects, help regulate autophagy, prevent premature apoptosis, and protect against various forms of cellular stress. Therefore, the goal presented by the authors is in line with the trend of modern medical research.

After a careful reading of the manuscript, I conclude that the abstract, introduction and remaining chapters comprehensively and adequately capture the issues discussed. The conclusions presented by the authors are consistent with the evidence and address the main research problem. In addition, I believe that the selection of bibliographic items is appropriate.

I recommend publication in its present form.

Reviewer 2 Report

In the current study, the authors evaluated the effects of hydrogen-rich water treatment on immobilization-induced skeletal muscle atrophy in a mouse model. Overall, it is an interesting and novel study that might be of interest to other muscle researchers. Nevertheless, this manuscript requires substantial improvements (see specific comments below).

Questions/ suggestions/ limitations of the study.

Introduction:

Line 57: “There are many circumstances such as fractures that lead to immobilization of body parts which result in muscle atrophy”. Please rewrite.

Line 57: “Several mechanisms have been suggested as possible mechanism of muscle atrophy”. Please rewrite.

Line 58: “higher production of oxidative stress in inactive muscles”. Please rewrite. Level?

Results:

Line 86: “After 7 days’ immobilization, the splints were removed…” I would start with the statement that the limbs were immobilized for 7 days and describe what happens. Please rewrite.

Figure 2. The statistical significance information on the differences between Control and Atrophy/H2 has to be presented in this Figure and Figure 3.

Figure 3. “P<0.05 compared to the atrophy group”. What * represents in Figure 2? What *** represents in Figure 3, Figure 4, Figure 6, and Figure 7?

Figure 4: What phase the cross-sectional area data represent: atrophy or recovery?

Figure 4: If you used microscope objective x10 then the overall magnification is x100.

Figure 5: The images in Figure 5 cannot be compared since the control is a cross-section and Atrophy and H2 are longitudinal/tangential sections. Please provide images of cross-sections of the whole GTN muscle done in the mid-belly region for all three experimental conditions. Indicate the magnification for these images. I would suggest the removal of this Figure since the data presented here cannot be trusted. Seven days of immobilization cannot produce such a large band of connective tissue in one localized area of the muscle section. This is most probably a defect of sectioning. Staining the perimysium of one of the fascicles is probably shown. 

Cross-section area data for Figure 4 are discussed in section 2.4. This should be discussed in the place where you discuss the rest of the data for Figure 4.

Line 156: “Results showed that after 7 days immobilization, serum troponin I level in atrophy group was significantly increased compared to control and that H2 treatment reduced it”. In which Figure?

Figure 7: Why the levels of expression of these genes are not shown for the control muscle? It will be low, but it is needed to see how much the atrophy increased the expression of these genes.

Discussion

The Discussion is very short and can be extended. The text should be improved.

Methods:

Please provide the Animal Protocol number and date of approval.

Please provide information on average water consumption in ml per mouse. Was there a difference in water consumption between regular and HRW-treated groups?

Line 286: “4.5. limb strength grip test” Needs a capital L at the beginning.

Line 297: A part of muscle samples from the hind limb were frozen in liquid nitrogen.” Please provide information on which part and how the uniformity of muscle sample collection was monitored.

Line 322: “specific qPCR Primers for target 322 genes (Macrogene Co, Seoul, South Korea)”. Please provide the catalog numbers and sequences of the primers.

The language of the manuscript also needs to be significantly improved. Many sentences are awkwardly structured and difficult to read and understand. 

Reviewer 3 Report

In the present paper Nazari et al describe the effects of hydrogen rich water in a model of muscle atrophy.

It provides a general evaluation of the effects of hydrogen by analyzing histological, biochemical and functional atrophy parameters.

Comments:

-       Authors should better develop the Introduction and the rationale of the study.

-       The images reporting H&E staining (Fig 4) are low quality, the presence of wide empty spaces suggests shrinkage artifacts. The Authors should insert a scale bar in the place of magnification.

-       In Fig 5 the control has been cut in the transversal plane, while Atrophy and H2 samples in the longitudinal plane. Authors should report similarly sectioned samples to allow an adequate comparison. Please insert the scale bar.

-       In my opinion, the functional measurement by the grip test is not appropriate. The model of atrophy consists in a unilateral hindlimb unloading/immobilization, therefore, it would be more appropriate to measure the functionality of the unloaded limb. An ex vivo contractile performance measurement would be more informative.

-       The Atrophy pathway should be better investigated.

English Language should be edited in several parts.